# Effects of Exercise and Sports Intervention and the Involvement Level on the Mineral Health of Different Bone Sites in the Leg, Hip, and Spine: A Systematic Review and Meta-Analysis

**DOI:** 10.3390/ijerph20156537

**Published:** 2023-08-07

**Authors:** Thiago P. Oliveira, Mário C. Espada, Danilo A. Massini, Ricardo A. M. Robalo, Tiago A. F. Almeida, Víctor Hernández-Beltrán, José M. Gamonales, Eliane A. Castro, Dalton M. Pessôa Filho

**Affiliations:** 1Graduate Programme in Human Development and Technology, São Paulo State University (UNESP), Rio Claro 13506-900, Brazil; airthiago@yahoo.com.br (T.P.O.); dmassini@hotmail.com (D.A.M.); tiagofalmeida.w@gmail.com (T.A.F.A.); eliane.castro@unesp.br (E.A.C.); 2Instituto Politécnico de Setúbal, Escola Superior de Educação, 2914-504 Setúbal, Portugal; mario.espada@ese.ips.pt (M.C.E.); ricardo.robalo@ese.ips.pt (R.A.M.R.); 3Life Quality Research Centre (LQRC-CIEQV, Leiria), Complexo Andaluz, Apartado, 2040-413 Rio Maior, Portugal; 4CIPER, Faculdade de Motricidade Humana, Universidade de Lisboa, 1499-002 Lisboa, Portugal; 5Department of Physical Education, School of Sciences (FC), São Paulo State University (UNESP), Bauru 17033-360, Brazil; 6Faculdade de Motricidade Humana, Universidade de Lisboa, 1499-002 Lisboa, Portugal; 7Research Group in Optimization of Training and Performance Sports, Faculty of Sport Science, University of Extremadura, 10005 Cáceres, Spain; vhernandpw@alumnos.unex.es (V.H.-B.); martingamonales@unex.es (J.M.G.); 8Faculty of Health Sciences, University of Francisco de Vitoria, 28223 Madrid, Spain; 9LFE Research Group, Universidad Politécnica de Madrid (UPM), 28040 Madrid, Spain

**Keywords:** sports, physical activity, bone, health

## Abstract

The current study analysed whether the osteogenic stimuli of exercises and sports have an independent effect on bone mineral density (BMD). Studies with a design having two different cohorts were searched and selected to distinguish the effect due to long-term involvement (i.e., athletes vs. non-active young with good bone health) and due to the planning of intervention (i.e., pre- vs. post-training) with exercises and sports. Moreover, only studies investigating the bone sites with a body-weight support function (i.e., lower limb, hip, and spine regions) were reviewed, since the osteogenic effects have incongruous results. A meta-analysis was performed following the recommendations of PRISMA. Heterogeneity (*I*^2^) was determined by combining Cochran’s Q test with the Higgins test, with a significance level of α = 0.05. The studies reporting the effect of involvement in exercise and sports showed high heterogeneity for the lower limb, total hip, and spine (*I*^2^ = 90.200%, 93.334%, and 95.168%, respectively, with *p* < 0.01) and the effect size on sports modalities (Hedge’s *g* = 1.529, 1.652, and 0.417, respectively, with *p* < 0.05) ranging from moderate to high. In turn, the studies reporting the effect of the intervention planning showed that there was no heterogeneity for the lower limb (*I*^2^ = 0.000%, *p* = 0.999) and spine (*I*^2^ = 77.863%, *p* = 0.000); however, for the hip, it was moderate (*I*^2^ = 49.432%, *p* = 0.054), with a low effect between the pre- and post-training moments presented only for the hip and spine (Hedge’s *g* = 0.313 and 0.353, respectively, with *p* < 0.05). The current analysis supported the effect of involvement in exercise and sports by evidencing the effect of either weight-bearing or non-weight-bearing movements on BMD at the femoral, pelvic, and lumbar bones sites of the athletes when comparing to non-athletes or non-active peers with healthy bones. Moreover, the effect of different exercise and sports interventions highlighted the alterations in the BMD in the spine bone sites, mainly with long-term protocols (~12 months) planned with a stimulus with high muscle tension. Therefore, exercise and sport (mainly systematic long-term practice) have the potential to increase the BMD of bones with body-weight support beyond the healthy values reached during life phases of youth and adulthood.

## 1. Introduction

Human bone is able to modify its structure according to different mechanical stimuli provided by involvement in exercise and sport, as well as other human motor actions, with an osteogenic effect able to elicit the increase in bone mineral content (BMC) and bone mineral density (BMD) [1,2]. Although involvement in exercise and sport is able to enhance bone mineral health, there are stimuli affecting local and systemic bone sites differently, which also differ regarding the magnitude of the osteogenic effects [3,4]. The magnitude of the osteogenic effect, according to the type of involvement in exercise and sport, was previously reported to be higher for the bone sites of the femoral neck, trochanter, and hip among elite athletes involved in terrestrial sports (soccer, field hockey, and volleyball) than athletes involved in aquatic modalities (swimming, artistic swimming, and water polo) of the same age group (21.5 ± 4.6 years). However, the BMC and BMD values of this latter group of athletes have been shown to be higher than those of the sedentary population, supporting the notion that the practice of water activities among elite athletes did not have a negative effect on BMD [5].

Thus, the magnitude of the effect provided by the mechanical stimuli on bones tends to be influenced by the environment of the practice. For example, another important condition is the weight-bearing mode of practice, which for Vlachopoulos et al. [6] was the factor influencing the higher BMC values in the hip, femur, and whole body among soccer players compared with cyclists and swimmers, which in turn did not differ with regard to BMC values in these same bone sites. This means that the mechanical stimuli differ according to the mode of practice of the exercise and sports (e.g., weight-bearing vs. non-weight-bearing practices), level of application (e.g., high vs. low impact), and magnitude of the load (e.g., muscle tension vs. cyclic movement) [1,3,6,7].

Also, the osteogenic stimuli from exercise and sport practice affect individuals differently, according to the fitness level, aging, lean mass, nutritional aspects, and endocrine responses [3,6,7]. For example, walking exercise planned for 20 weeks (combined or not with other exercises) increased BMD in the femoral and lumbar region (~2 to 5%) among older and sedentary individuals [8], while no osteogenic effects at the lumbar spine, femoral neck, and pelvis were observed in trained male and female runners compared to active individuals [5,9,10].

The hip and spine regions are bone sites with clinical relevance in preventing bone frailty, compromising health status by decreasing motor performance and independent locomotion [11]. There is evidence suggesting that chronic involvement in resistance training combined with jumps promotes the increase in the BMD of the whole body, the lumbar vertebrae, and the hip in active men (aged 44 ± 2 years) with previous osteopenia at these bone sites [12,13]. Indeed, exercise and sports practices enabling changes in body composition favouring an increase in muscle mass and strength have a positive effect on BMD in different bone sites, revealing that the mechanostat response of bone might be enhanced and spread to bone sites directly and not directly involved in movement [14,15], which is an assumption aligned with the notion of lean mass, and muscle size and strength have been shown to influence bone BMD whatever the age group, sex, and race [14,16,17].

While the mechanostat is an assumption associating muscle tensional stimuli to bone mineral deposition and morphology, possibly through modulation of osteoblastic activity [18,19], there is also the interaction between muscle and bone support for paracrine and endocrine factors that can stimulate osteogenesis at local and systematic levels [20,21]. For example, the serum and muscle-derived levels of insulin-like growth factor (IGF)-1 can modulate different myokines and osteokines that mediate anabolic responses (e.g., increase mass and strength of muscle and bone) and that are also involved in mechanical signal translation into biological events [10,21]. Although exercise and sports play a key role in the magnitude of the mechanical stimuli on bone, the conditions of practice able to improve muscle–bone interactions require further investigation.

Hence, if both exercise and sport practice can provide stimuli able to enhance the BMD in bone sites that were usually susceptible to pathologies (e.g., femur, hip, lumbar vertebrae); then, the characteristics of exercise and sport (e.g., mode of practice, level of involvement, and planning), which might optimize such an effect of the mechanical stimuli on bone, still need to be gathered and analysed to demonstrate the evidence. The current proposal hypothesized that exercise and sport involvement (i.e., mainly those practiced in weight-bearing mode) provides information regarding the effectiveness of these practices to improve bone health at these anatomical sites. The relevance in highlighting the magnitude of the osteogenic effect from different sport modalities in bone sites susceptible to frailty, is to support strategies towards the prevention of bone injuries among non-atheletes and athletes during sport practice, as well as to improve bone health in the elderly and enhance bone development during growth [1,9,10,22].

Thus, this systematic review aimed to analyse the effect of different exercise and sport involvements on the BMD in lower limb, hip, and spine bone sites, which are specific sites susceptible to frailty and affected by osteoporosis. For this purpose, a meta-analysis was performed, encompassing cross-sectional studies analysing whether the osteogenic effect is higher for a given level of exercise and sport practice when athletes (i.e., high involvement) are compared to the healthy bone non-athlete population (i.e., low involvement), as well as longitudinal studies analysing the evidence of osteogenic effects with systematized practice of exercise and sports during a given time course

## 2. Materials and Methods

### 2.1. Search Strategy

This meta-analysis followed the recommendations of PRISMA (Preferred Reporting Items for Systematic Reviews and Meta-Analyses) [23] (see Appendix A) and received the registration number CRD42021251870 in PROSPERO baseline records. The computerized searches were carried out between the 5th and 12th of December 2022 in the VHL Regional Portal, PEDro, PubMed, SciELO, ScienceDirect, and Web of Science electronic databases (see Appendix A). The high-sensitivity search was elaborated using descriptors according to the population, intervention, comparison, and outcome (PICO) strategy: Populations: “Health” OR “Population Health”; Intervention: “Sports” OR “Exercise”; Comparator: Modalities vs. controls (involvement with exercise and sport); pre- vs. post-training (interventions planed with exercise and sport); Outcomes: Bone Mineral Density” OR “Bone Mineral Content.

Searches were also carried out in the references and citations from the eligible articles to add relevant titles. In addition, citation tracking of the included studies was carried out through the Pubmed, Scopus, and Google Scholar databases. Gray literature (e.g., abstracts, conference proceedings, editorials, dissertations, and theses) was not included. Finally, attempts were made to contact the authors of the selected articles via e-mail to request any lack of relevant information. To avoid selection bias, two authors performed the searches (D.A.M. and T.P.O.). After conducting the research, the authors compared the lists of included and excluded studies, and the observed discrepancies were analysed through discussion and agreement with other co-authors (D.M.P.F., E.A.C., and T.A.F.A.).

### 2.2. Inclusion and Exclusion Criteria for the Studies

Studies that presented quantification of BMD were included. The inclusion criteria adopted were (i) complete studies carried out in humans between 18 and 45 years of age; (ii) that quantified bone metabolism by the double-beam X-ray (DXA) method; (iii) peer-reviewed studies published in English; and (iv) published in the last five years (preferably). The exclusion criteria adopted were (i) studies carried out in clinical populations that interfere with osteogenic metabolism [24]; (ii) that administered osteogenic supplements or medication; (iii) case studies and literature reviews (systematic review and meta-analysis); and (iv) studies published in other languages.

### 2.3. Data Extraction

The main characteristics of the selected studies are found in Table 1 (level of involvement with exercise and sport) and Table 2 (interventions planned with exercise and sport). The data were extracted by several authors (T.P.O., M.C.E., F.J.S., T.A.F.A., E.A.C., and D.M.P.F.) using a pre-pilot spreadsheet and independently verified by a different author (D.A.M.) from the review team. The following data were extracted: (i) authors’ names, (ii) year of publication, (iii) population characteristics (sample size, sex, age, height, and body mass), (iv) sports modalities, (v) BMD measures (region and pre- and post-training values).

### 2.4. Assessment of Methodological Quality and Risk of Bias

This procedure was performed by two authors (D.A.M. and T.P.O.), and it was carried out using the PEDro Scale of 11 points (Physiotherapy Evidence Database) (see Appendix A), which assigns one point to the study if the criterion was met or 0 if not [25]. These researchers were well familiarized with the classification of the studies’ quality for systematic analysis. As criterion 1 concerns external validity, this was not considered in the total score; in the same way, criteria 5, 6, and 7 were removed due to the impossibility in studies of interventions with physical exercise to allocate the groups of the participants blindly, as researchers rarely act blindly [18]. With the removal of these items, the maximum value of the PEDro scale is seven points, with adjusted ratings ranging from 0–3 being “poor quality”, 4 being “moderate quality”, 5 being “good quality”, and 6 to 7 being “excellent” quality” [26,27]. Studies with poor methodological quality were excluded from this meta-regression analysis [26].

### 2.5. Statistical Analysis

The statistical analysis was performed by an author (D.A.M.) and reviewed by another (D.M.P.F.). The studies were initially organized as cross-sectional (analysing involvement) and longitudinal (analysing intervention) to the analysis of the effect on BMD from data reported for exercise and sport involvement and programs of experimental intervention, respectively. The magnitude of the results for each study was then determined by Hedge’s *g* and 95% confidence interval (CI_95%_) for the different bone sites in the lower limb, hip, and spine in each exercise or sport condition (involvement or intervention) of the studies included in the meta-analysis. For these estimates, the sample size and the mean and standard deviation of BMD values in the cross-sectional (athletes vs. healthy bone young individuals) and longitudinal (alterations between pre- and post-intervention) were used for each bone site of the lower limb, hip, and spine investigated in each study. The effect size for Hedge’s *g* was considered <0.19 (trivial), 0.20–0.49 (small), 0.50–0.79 (medium), 0.80–1.29 (large), and >1.30 (very large) [28]. Heterogeneity (*I*^2^) was determined by combining Cochran’s Q test with the Higgins and Thompson test [29], whereby the values are categorized as: 0 < *I*^2^ ≤ 25% “non-heterogeneity”; 25% < *I*^2^ ≤ 50% “low heterogeneity”; 50% < *I*^2^ ≤ 75% “moderate heterogeneity”; and >75% “high heterogeneity” between studies [30]. A fixed-effect model was employed in the absence of inconsistency (*I*^2^ ≤ 25%). The bias analysis could not be performed, since less than 10 studies were included for each cohort (cross-sectional and longitudinal) of this meta-analysis [31]. For all analyses, a significance level of α = 0.05 was adopted.

## 3. Results

Figure 1 shows the flowchart of the systematic review and meta-analysis. Fifteen studies (eight cross-sectional and seven longitudinal cohorts) were selected independently by five reviewers, being published in European countries (36.4%) and in the United States of America (63.6%). This selection included 869 participants (57.3% women and 41.0% men), with an average age between 25.3 ± 5.7 years and 23.9 ± 5.6 years. The average methodological quality of the studies corresponded to 4.25 for cross-sectionals and 4.85 for longitudinal (Table 1 and Table 2), and all studies being considered moderate regarding the methodological quality. Table 1 presents the characteristics of the cross-sectional studies. All the participants involved with exercise and sport were athletes (e.g., individuals engaged with regular training for at least two years, and competing regularly) in the studies shown in Table 1, whereas the participants under intervention planned with exercise and/or sport (Table 2) were active individuals (non-athletes) or athletes. The exercises and sports included in Table 1 were mostly weight-bearing activities (i.e., nine different modalities), with the exception of three modalities which were non-weight-bearing activities practiced in water. With regard to the intensity, the weight-bearing activities were considered moderate to high in terms of bone-loading forces, while the non-weight-bearing activities were typical exercises or sports with high muscle tension requirement. The main characteristics of the exercises and sports included in the intervention planning are shown in Table 2.

For the lower limb bone sites (Figure 2A), the effect of the involvement with exercise and sports modalities was presented in relation to the control groups (g = 1.529, CI_95%_: 1.150–1.908, *p* < 0.001 (very large)) under conditions of high heterogeneity (*I*^2^ = 90.2%, Q_[21]_ = 231.272, *p* < 0.001) and also based on the random model. The same is true for the hip bone sites (Figure 2, Panel B), where the effect of sports modalities on the control groups (g = 1.652, CI_95%_: 1.060–2.244, *p* < 0.001 (very large)) is also observed in conditions of high heterogeneity (*I*^2^ = 93.344%, Q_[11]_ = 165.254, *p* < 0.001) and also based on the random model. Finally, the spine bone sites (Figure 2C) also demonstrated the effect of sports modalities compared to the control groups (g = 0.417, CI_95%_: −0.275–1.108, *p* = 0.237 (small)) under conditions of high heterogeneity (*I*^2^ = 95.168%, Q_[10]_ = 206.955, *p* < 0.001) and based on the random model.

Table 2 presents the characteristics of the longitudinal studies. For the lower limb bone sites (Figure 3A), in the lack of heterogeneity (*I*^2^ = 0.000%, Q_[18]_ = 4.972, *p* = 0.999) and based on the fixed model, there was no effect between the pre- and post-training moments (Hedges’ *g* = −0.010, CI_95%_: −0.175–0.156, *p* = 0.910 (trivial)). Likewise, the hip bone sites (Figure 3B) under moderate heterogeneity (*I*^2^ = 49.432%, Q_[7]_ = 13.843, *p* = 0.054) and based on the random model, presented no effect between the pre- and post-training (Hedges’ *g* = 0.313, CI_95%_: −0.023–0.650, *p* = 0.068 (small)). In contrast, the spine bone sites (Figure 3C) showed heterogeneity (*I*^2^ = 77.863%, Q_[7]_ = 31.621, *p* = 0.000) and based on the random model have a significant effect between the pre- and post-training moments (Hedges’ *g* = 0.353, CI_95%_: −0.162–0.869, *p* = 0.179 (small)).
ijerph-20-06537-t001_Table 1Table 1Main characteristics of the selected cross-sectional studies concerning population characteristic, exercise or sport effect on BMD, and quality analysis.StudyParticipantsExercise/Sport (Bone Site)BMD (g/cm^2^)Methodological QualityAthleteControlPointsClassificationBellveret al.                      204 elite female athletesSwimming (FN)0.994 ± 0.1000.903 ± 0.1404and 126 controlsSynchronized Swimming (FN)1.103 ± 0.0900.903 ± 0.140Moderate(21.5 ± 4.6 years).Water Polo (FN)1.172 ± 0.1200.903 ± 0.140quality
Soccer (FN)1.240 ± 0.1400.903 ± 0.140

Field Hockey (FN)1.155 ± 0.1100.903 ± 0.140

Volleyball (FN)1.272 ± 0.1400.903 ± 0.140

Swimming (Tr)0.811 ± 0.080.677 ± 0.13

Synchronized Swimming (Tr)0.865 ± 0.110.677 ± 0.13

Water Polo (Tr)0.889 ± 0.080.677 ± 0.13

Soccer (Tr)1.039 ± 0.140.677 ± 0.13

Field Hockey (Tr)1.030 ± 0.090.677 ± 0.13

Volleyball (Tr)1.048 ± 0.110.677 ± 0.13

Swimming (P)1.019 ± 0.1100.924 ± 0.100

Synchronized Swimming (P)0.991 ± 0.1000.924 ± 0.100

Water Polo (P)1.149 ± 0.0600.924 ± 0.100

Soccer (P)1.231 ± 0.1300.924 ± 0.100

Field Hockey (P)1.185 ± 0.1600.924 ± 0.100

Volleyball (P)1.184 ± 0.1200.924 ± 0.100

Swimming (L3–L4)1.161 ± 0.1401.057 ± 0.160

Synchronized Swimming (L3–L4)1.107 ± 0.1101.057 ± 0.160

Water Polo (L3–L4)1.265 ± 0.0901.057 ± 0.160

Soccer (L3–L4)1.341 ± 0.1601.057 ± 0.160

Field Hockey (L3–L4)1.258 ± 0.1001.057 ± 0.160

Volleyball (L3–L4)1.431 ± 0.1801.057 ± 0.160
Lees et al.26 elite male fast bowlers and 26Cricket Fast bowlers (FN)2.138 ± 0.1851.715 ± 0.2324
normally active controlsCricket Fast bowlers (Tr)1.811 ± 0.1611.469 ± 0.219Moderate
(24.3 ± 4.2 years)


qualityTam15 elite male Kenyan runnersRunning (FN)0.945 ± 0.1660.927 ± 0.1354et al.(24.4 ± 4.7 years) and 23 controlsRunning (PF)1.265 ± 0.1841.074 ± 0.145Moderate
(29.0 ± 4.1 years)Running (LS)1.009 ± 0.1661.040 ± 0.116qualityHind52 male rugby playersRugby (FN)1.325 ± 0.2001.178 ± 0.2004et al.(26.6 ± 4.4 years) and 32Rugby (TH)1.442 ± 0.2001.195 ± 0.200Moderate
controls (25.0 ± 3.9 years).


qualityPiasecki15 female long-distance runnersRunning (P)1.140 ± 0.0201.11 ± 0.015et al.(eumenorrheic) and non-athleticRunning (L3–L4)1.16 0± 0.0301.19 ± 0.03Good
controls (*n* = 15) (17 ± 42 years)


qualityBolam30 male boxers (30.1 ± 6.4 years),Boxing (TH)1.045 ± 0.1341.059 ± 0.1245et al.and 32 non-boxing activeBoxing (LS)1.131 ± 0.1281.131 ± 0.124Good
controls (30.7 ± 6.1 years)


qualityMcCormack60 runners (age): 27 male (age 19.7Male Runners (FN)0.934 ± 0.0310.866 ± 0.2804et al.± 1.2) and 33 female (age 20.3 ±Female Runners (FN)0.921 ± 0.0240.910 ± 0.300Moderate
1.8); 47 Control: 23 male (age 20.0Male Runners (TH)1.062 ± 0.030.959 ± 0.028quality
± 0.8) and 24 female (age 19.8 ± 0.6)Female Runners (TH)1.039 ± 0.0241.024 ± 0.03


Male Runners (S-M)0.912 ± 0.0290.923 ± 0.026


Female Runners (S-W)1.002 ± 0.0231.046 ± 0.028
Sagayama33 college athletes (aged 18 ± 22)Wrestler (Legs)1.422 ± 0.091.279 ± 0.1004et al.years): 11 male wrestlers, 9 judo,Judo (Legs)1.346 ± 0.0861.279 ± 0.100Moderate
13 endurance athletes,Endurance Exercise (Legs)1.262 ± 0.0971.279 ± 0.100quality
and 8 control



BMD: Bone Mineral Density, FN: Femoral Neck, Tr: Trochanter, P: Pelvis, TH: Total Hip, LS: Lumbar Spine, L3–4: Lumbar Spine at 3–4th vertebrae. The control individuals which were compared with athletes showed the following characteristics: Bellver et al. (2019)—sedentary controls with no formal training in sport, and with no regular exercise practice; Lees et al. (2017)—recreationally active individuals (e.g., <3 sports-specific sessions per week); Tam et al. (2017)—sedentary healthy male individuals from different ethnicities; Hind et al. (2015)—male healthy sedentary individuals; Piasecki et al. (2018)—female non-athletic individuals performing exercise with <2 h per week (all participants declared the phase of menstrual cycle and contraceptive method); and Bolam et al. (2016)—male healthy individuals with no standard training planning (e.g., <2 exercise sessions) per week.
ijerph-20-06537-t002_Table 2Table 2Characteristics of the selected longitudinal studies concerning population, exercise or sports effects on BMD, and quality analysis.StudyParticipants Exercises/Sport (Bone Site)BMD (g/cm^2^)Methodological Quality Pre-TrainingPost-TrainingPointsClassification Caruso et al.            13 healthy subjects, 2 men and 11 women(29.4 ± 12 years).Study Time:30training sessions in 70 ± 6.3 days (range 58–84 days.  30 healthy women(22 ± 2 years). Study Time:12 weeksInertial Exercise Trainer (FN-Le)Inertial Exercise Trainer (FN-Ri)Inertial Exercise Trainer (Tr-Le)Inertial Exercise Trainer (Tr-Ri)Inertial Exercise Trainer (ITr-Le)Inertial Exercise Trainer (ITr-Ri)Inertial Exercise Trainer (DF-Le)Inertial Exercise Trainer (DF-Ri)Inertial Exercise Trainer (W-Le)Inertial Exercise Trainer (W-Ri)Inertial Exercise Trainer (TH-Le)Inertial Exercise Trainer (TH-Ri)0.990 ± 0.1800.990 ± 0.1100.810 ± 0.1100.800 ± 0.0901.270 ± 0.1801.240 ± 0.1101.190 ± 0.1101.180 ± 0.1801.000 ± 0.3001.000 ± 0.3001.070 ± 0.1101.060 ± 0.1101.010 ± 0.1800.990 ± 0.1800.820 ± 0.0900.810 ± 0.0901.270 ± 0.1801.270 ± 0.1101.220 ± 0.1801.160 ± 0.1801.000 ± 0.2001.000 ± 0.3001.080 ± 0.1101.070 ± 0.1105Goodquality         Mostiet al.14 healthy male professional soccer players(17.5 ± 0.8 years).Study Time:27 weeksHA strength training (FN)HA, maximal strength training (Tr)HA maximal strength training (ITr)HA, maximal strength training (TH)HA maximal strength training (LS)0.851 ± 0.0860.747 ± 0.0541.149 ± 0.1030.977 ± 0.0811.023 ± 0.0730.863 ± 0.0920.747 ± 0.0591.138 ± 0.1000.967 ± 0.0791.002 ± 0.0814Moderate qualitySuarez-Arroneset al.EOT in elite soccer players (Leg-Le)EOT in elite soccer players (Leg-Ri)EOT in elite soccer players (P)EOT in elite soccer players (TS)1.410 ± 0.0801.400 ± 0.0901.600 ± 0.2101.230 ± 0.2201.380 ± 0.0801.360 ± 0.0801.540 ± 0.2101.220 ± 0.1905GoodqualityFeitoet al.     Kurganet al.    Fristrupet al.  26 recreationally active adult men (*n* = 9, 34.2 ± 9.1 years) and women (*n* = 17, 36.4 ± 7.9 years).Study Time:16 weeks  15 elite women heavyweight rowers(27.0 ± 0.8 years).Study Time:42 weeks  28 RecreationallyHandball training14 men and 14 womenage (24.1 ± 2.6 years).Study Time: 12 weeksHigh Intensity functional training: Multimodal exercises (Leg-W)High Intensity functional training: Multimodal exercises (Leg-M)  Heavy-weight row (P) Heavy-weight row (LS)    Recreationally Handballtraining (TH)    1.250 ± 0.090   1.530 ± 0.090   1.260 ± 0.030 1.230 ± 0.030    1.099 ± 0.115 1.250 ± 0.090   1.510 ± 0.080   1.220 ± 0.030 1.200 ± 0.030    1.117 ± 0.115 4Moderate quality   5Goodquality    7Excellentquality Infantino et al.21 male and18 distance femalesRunners(19.5 ± 0.8 years).Study Time:12 monthsDR (TH-M)DR (TH-F)DR (FN-M)DR (FN-F)DR (Sl-M)DR (Sl-F)DR (Spa-M)DR (Spa-F)1.065 ± 0.0371.057 ± 0.0330.944 ± 0.0390.930 ± 0.0350.783 ± 0.0250.736 ± 0.0220.956 ± 0.0290.970 ± 0.0261.089± 0.0351.037 ± 0.0320.942 ± 0.0360.919 ± 0.0320.817 ± 0.0240.723 ± 0.0220.984 ± 0.0280.955 ± 0.0254ModeratequalityHA, High acceleration; EOT, Eccentric-overload training. BMD: Bone Mineral Density, FN: Femoral Neck, Tr: Trochanter, Le: Left, Ri: Right, ITr: Inter Trochanter, DF: Distal Femur, W: Ward Triangle, P: Pelvis, TH: Total Hip, LS: Lumbar Spine, Leg-W: Leg Women, Leg-M: Leg Men, TS: Thoracic Spine, P: Pelvis, LS: Lumbar Spine, Spa: Anterior-posterior spine, Sl: Lateral spine, DR: Distance running, M: Male, and F: Female.

## 4. Discussion

The osteogenic stimulus observed with exercise and sport involvement showed an effect on the bone sites of the lower limbs, hip, and spine. Such an inference was obtained by analysing both the level of involvement and intervention planning with exercise and sports, respectively, in the cross-sectional and longitudinal cohort studies. Regarding the level of involvement, there is wide homogeneity regarding the osteogenic effect of exercise and/or sports when compared to the lack of stimulation, such as physical inactivity, especially on the bone sites of the lower limbs, hips, and spine (Figure 1A–C). In contrast, when considering the results of the intervention planning that provide an analysis of the BMD response during a given period (i.e., training protocols for ~10 to 42 weeks), the studies reported changes with an insufficient statistical level for the variations observed in bone sites highlighted in the present study. However, in these studies, the osteogenic effect of some exercises and sports, such as soccer, rowing, and strength training (i.e., performed with speed), stands out in spine bone regions.

Furthermore, the present study corroborates the assumption postulating that there is neither an ideal exercise or sports practice able to promote higher bone mass than another, or to target a specific bone site [1,14]. However, the present study presented evidence that exercise and sports enabling weight-bearing practice are effective to increase bone mass. Therefore, it is recommended to be involved with exercise and sport practice, in which appropriate control of training loads is ensured, for the enhancement of bone health.

### 4.1. Evidence of BMD Alteration According to Exercise and Sport Involvement

The level of involvement with exercise and sport showed to have an important effect on the BMD of the lower limbs, hips, and lumbar vertebrae when athletes (i.e., high level of involvement) were compared to sedentary or non-athlete peers assigned to the control group (Figure 1A–C). The different responses of BMD also stand out when comparing the sports modalities with regard to the magnitude of the osteogenic stimuli of their practice (Figure 1A–C). From the studies analysed, the reported findings conclude that exercise and sport practice with impact mode of stimulus are more osteogenic than low-impact activities (e.g., cycling) or with no impact (e.g., swimming) [32,33,34]. However, in the specific case of swimming, there are studies showing that swimming athletes can have lower BMD values than non-athletes, while other studies showed no significant differences when compared to non-athletes [33,34,35,36,37,38]. According to the current analysis, it was evident that aquatic activities have a low effect on BMD only if compared to the level of stimuli of impact from terrestrial weight-bearing activities.

Indeed, the results of Bellver et al. [5] for water (i.e., swimming, synchronized swimming, and water polo) and land sports (i.e., soccer, hockey, and volleyball) practices showed greater effectiveness of osteogenic stimulus in favour of sports practice on land for all bone sites studied. However, water sports showed higher BMD values compared to the control group (sedentary) for the femur (1.085 ± 0.12 vs. 0.903 ± 0.14 g/cm^2^), trochanter (0.854 ± 0.99 vs. 0.677 ± 0.13 g/cm^2^), hip (1.034 ± 0.97 vs. 0.901 ± 0.15 g/cm^2^), and spine L1-L4 (1.166 ± 0.13 vs. 1.057 ± 0.16 g/cm^2^). Indeed, athletes involved with high impact stimuli, such as fast cricket players, showed high BMD values in different sites of the femur when comparing with active young peers [39].

However, there are conflicting results regarding the superior effectiveness of the impact stimulus compared to the other types of stimuli (i.e., high muscle tension) when different types of exercises are compared. For example, Ubago-Guisado et al. [40] compared the effect of practicing Zumba^®^ (i.e., high-impact terrestrial activity) with the practice of Aquagym (i.e., low-impact aquatic activity), both prescribed three times a week for a period of 12 weeks, involving only sedentary women aged 30–50 years. No differences were found for BMC and BMD, suggesting that the stimulus for bone mass is similar in bone sites such as whole body (excluding the head value), lumbar vertebrae, and right hip.

Moreover, when comparing individuals involved in exercise with sedentary peers, the effectiveness of the osteogenic stimulus can be distinguished regardless of the mode of stimuli. For example, Ubago-Guisado et al. [40] observed high BMD values of the hip for participants performing a high-impact condition (Zumba^®^) or low-impact aquatic activity (Aquagym) when compared to sedentary peers. This fact allowed the aforementioned authors to conclude that both activities are recommended to reduce the progressive fragility of bone mass in previously inactive women.

For this reason, even though sports modalities with low-impact intensity (including those performed in an aquatic environment) present controversial results regarding the magnitude of the osteogenic effect (Figure 1A–C), the regularity of the practice is important to the occurrence of the osteogenic effects. For example, studies comparing BMD responses in water sports athletes to the control condition (in which the individual is not submitted to any type of training) reported the effect on the BMD of the lumbar vertebrae in woman at the proximal portion [5]. It is noteworthy that these authors involved women athletes of Olympic level; therefore, confounding effects of the training status due to the high volume of weekly training (12–15 h), as well as other activities performed outside the aquatic environment and with external impact overload, may have offered an additional osteogenic stimulus.

In addition, there are studies that have observed null osteogenic effects even with the practice of high-impact sports in a terrestrial environment. In the present study, four cross-sectional studies supported this fact (Figure 1B,C). For example, for Bolam et al. [41], the bone sites of the hip and lumbar vertebrae did not demonstrate an osteogenic effect in fighting activities or racket sports. However, it is important to note in this study that the BMD of amateur boxers was compared with the BMD of active volunteers (i.e., non-boxers), which could be engaged in other sports activities, including resistance training with a similar training load ratio, which might provide similar or higher osteogenic stimuli. Another example was the null effect of involvement with Wrestling, Judo, and Endurance Exercises on the BMD of legs in male young college athletes [42]. Regarding these conditions, the absence of specific effects in an experimental group is expected, demonstrating that mechanical stress favouring bone remodelling has a limit of normality for BMC/BMD in adult individuals enrolled in regular sports training programs [10,43,44]. Indeed, the increment in BMD might reach a set-point for the osteogenic effect mediated to the exercises and/or sports [9]. Thus, while the specificity of the osteogenic stimulus on BMD is different between sports, the effect has a limit related to the conditioning level, biological maturation, sports experience, lean mass, nutritional aspects, and endocrine responses [9,44].

Other studies, such as those of Tam et al. [44], Piasecki et al. [10], and McCormack et al. [45] also showed a lack of osteogenic effect for the practice of running in the bone sites of the lumbar vertebrae, femoral neck, and spine, although running is considered an osteogenic modality [46]. However, according to Piasecki et al. [10], the lumbar vertebrae is considered a bone site that does not receive a great mechanical load during running, due to the mechanical cushioning of several muscles adjacent to the site, which therefore can be also linked to the notion of site-specific effect according to the mode of stimuli during exercise and sport involvement [14]. This is an effect also observed by Hind et al. [22] who reported lower BMD values for the lumbar vertebrae compared to the hip region among runners. For these authors, the greater load magnitude for the hip region instead of the lumbar vertebrae may cause this difference during running.

Another aspect that may be related to the lack of effect for running activity is the low BMD value found in some long-distance runners, mainly women. Dengel et al. [47], in a comparison between different athletics modalities (i.e., pole vault, sprinting, jumping, medium- and long-distance running, and multiple events), observed that both men (1.25 ± 0.10 g/cm^2^) and women (1.16 ± 0.09 g/cm^2^) had lower BMD values compared to the other studied sports modalities. The lower BMD found in this population of athletes may be due to the relative energy deficiency with such a sport practice, which is an inadequate availability of energy to meet the metabolic demands of the body during sport practice that can also compromise endocrine secretion and thus account for the low BMD mainly for lumbar vertebrae [48,49]. Therefore, metabolic disorders with overtraining might be considered a negative factor for bone health derived from exercise and sport involvement.

However, the current meta-analysis has shown the osteogenic effectiveness of the exercise and sports modalities involvement on bone sites of the lower limbs (Hedge’ *g* = 1.529; *p* < 0.001), spine (*g* = 0.417; *p* = 0.237), and hip (*g* = 1.652; *p* < 0.001). Hence, training is highly recommended to strengthen bone sites supporting body weight, but athletes’ lifestyle involves planning, nutritional, and medical periodic evaluations. 

### 4.2. Evidence of BMD Alteration after Different Interventions Planned with Exercise and Sport

The revised studies did not show an important osteogenic effect in the lower limb and hip bone sites when comparing BMD values between pre- and post-intervention planned with different exercises and sports. The effect size (Hedge’s *g* in Figure 2A,B) was considered trivial, i.e., too small, for the bone sites of lower limb and hip (Figure 2A,B). However, the spine bone sites showed an osteogenic effect on BMD values in the pre- and post-intervention moments, with an effect size considered small (Figure 2C).

In Figure 2C, however, the studies of Kurgan et al. [50] and Infantino et al. [51] evidenced significant and large effect size (g = 0.973; *p* = 0.010) for the increase of BMD in the pelvis and lumbar spine bone sites. Interestingly, the relative weight of this study (24.19) was similar to the other studies (24.19 ± 0.24), which would indicate that the results presented by this study did not influence the significance or the effect size found for these bone sites after intervention. Despite the bone sites of the lower limb and hip showing no osteogenic effect between the pre- and post-training moments, it is important to highlight the alterations of the BMD in lower limbs reported by Suare-Arrones et al. [52], as well as in the hip, also reported by Suare-Arrones et al. [52]. Again, these results were not noticeable to influence overall effect size of the different interventions, but evidence a site-specific effect of long-term resistance and running training (respectively) on lower limbs, hip, and spine.

The first hypothesis for the lack of an overall large effect size might be related to the time, questioning whether the interventions lasting a satisfactory time to ensure bone tissue modifications parallel to that commonly reported for skeletal muscle tissue [53,54]. Thus, even though exercise/sport can stimulate bone structure, a long period of time is required for bone remodelling to be consolidated. In fact, this analysis is aligned to the statement that the increase in bone resorption takes about 3 to 4 months to complete a remodelling cycle (bone resorption, formation, and mineralization), and takes at least 6 to 8 months to achieve a new bone mass in a state that is minimally measurable [55]. The second hypothesis is related to the state of bone maturity of the participants at the beginning of the intervention, since some individuals could have acquired a satisfactory bone mass prior to the intervention, given their sports/osteogenic experience, thus hindering additional gains that can be observed with the intervention (i.e., the set-point statement [43]).

Indeed, among the revised studies, there is heterogeneity in terms of the duration for each intervention, and also the levels of physical fitness differed between the participants, requiring a particular analysis of each study. With regard to the duration of the intervention, the studies planning short-term training (four studies) involved periods of 70 ± 6.3 days [56], 12 weeks [57], 16 weeks [58], and 12 weeks [51], and all showed no significant results. For Mosti et al. [56], the increase of 2.2% in the lumbar vertebrae and 1% in the hip bone revealed differences only within the intervention group, which included strength training, but the comparison to the control group (performing no exercise or sport) showed no differences. However, it is important to consider that the study of Mosti et al. [57] included healthy women aged 22 ± 2 years, suggesting that the lack of differences might be influenced by the bone development in this age group, as well as to possible previous experience with osteogenic stimuli, such as exercise practice. However, the effect size for this study on the spine site showed *g* = 0.264, which is considered small. 

Likewise, the study of Caruso et al. [56], which employed 30 inertial exercises for 70 ± 6 days in 13 volunteers (2 men and 11 women) with no previous experience with such an exercise, found no differences for the BMD values of the hip and legs between the pre- and post-training moments, when an active limb was compared to the control limb. Interestingly, despite the short-training period, these authors found a significant increase in BMD of the heel bone and also a significant reduction (386.2 ± 97 to 336.9 ± 86 pg·mL^−1^) of the type 1 collagen C-terminal telopeptide marker, which presented a negative relationship with bone BMD in lower limbs. Therefore, the intensity of the tensional stimuli, like that provided by resistance training, is effective for short-term BMD responses in both sexes. 

Concerning the Feito et al. [58] study, which is the only study planned with 16 weeks (i.e., short-training intervention), involving nine men (34.2 ± 9.1 years) and 17 women (36.4 ± 7.9) recreationally active, observed no changes for the pre- vs. post-intervention in legs, lumbar vertebrae, and pelvis BMD, regardless of sex. However, the authors reported that full-body BMD changed by 1.4 ± 4.4%, which allowed the authors to conclude that 16 weeks of high-intensity functional training was also effective in promoting an increase in muscle strength, body composition, and bone health. Contradictorily, there are findings reporting no effect of resistance training on BMD in adults with longer interventions [59]. Moreover, the null effect of high impact stimulus was observed after Handball training in pelvis and lumbar spine either in male or female young players, which was accounted to the short period of intervention [60].

Such controversial reports on the unfavourable effect of exercise and sport interventions on BMD values [56,57,58,60] corroborates the statement that a long duration is required for the mineral remodelling to take place in response to a given osteogenic mechanical stimuli. In addition, BMD remodelling is also affected by the pre-intervention involvement with exercise and sport, which might compromise the effectiveness of a given planning of intervention to evidence improvements of BMD. In contrast, the studies reporting favourable results on BMD [61,62] consolidated the effectiveness of resistance training and weight-bearing to stimulate the biological mechanisms responsible for improving bone mass and strength [59]. Moreover, all analysed interventions did not report BMD reduction after a given period of time, evidencing the important role on bone health maintenance, which is a considerable factor when the risk of bone frailty and pathologies aimed to be prevented.

Finally, the interventions planned with long durations, such as Suarez-Arrones et al. [52] and Kurgan et al. [50] with approximately nine months, observed improvement of the whole-body BMD, as well as for the legs, hip, and spine bone sites of highly trained athletes (e.g., professional Italian young male soccer players (17.5 ± 0.8 years), and for young (27.0 ± 0.8 years) Olympic rowers, respectively). These studies were highlighted due to the evidence of the osteogenic effects on bone sites of the lower limbs, hip, and spine with long-term interventions in individuals with a high level of physical fitness and involvement with the activity, despite the results of this meta-analysis not indicating a positive effect of exercise/sport on BMD in the lower limb (Hedges’ *g* = 0.010; *p* = 0.910), and hip (*g* = 0.313; *p* = 0.068). These results corroborate that BMD increase is also dependent on other non-mechanical stimuli, such as biological maturation, aging, lean mass, nutrition, and endocrine responses (IGF-1, leptin, testosterone) [52,61,62,63,64,65,66].

### 4.3. Limitations

The limitation of the study, both in the cross-sectional and longitudinal cohorts, was the small number of sports modalities included and the impossibility of subgroup analysis and meta-regression by modalities and participants’ characteristics (i.e., training status, sex, race, and age). Another important concern is the lack of information about supplementation and drug use among the participants, which might influence BMD values (mainly the athletes) due to the role of sex-specific humeral secretion on bone metabolism modulation [20,21]. Moreover, sex and age both have an influence on BMD values due to the differences in body composition and humeral serum levels when comparing men vs. women and young vs. elderly individuals [7,14,16,18]. Thus, these are additional limitations of the current study, since the differences between athletes from different sexes and age group regarding BMD value would be important information to address the role of each exercise and sport as independent factors with influence on healthy bone mass. Therefore, future studies should analyse different sports modalities and characteristics of the sample population, as well as verify the effect of these sports modalities in individuals with bone metabolism disorders (i.e., obese, diabetic, abnormal menstruation, osteopenic, and osteoporotic individuals), as well as the effectiveness of the osteogenic stimuli from a given exercise or sport practice on the BMD of athletes differing with regard to the race, sex, and the level of training.

## 5. Conclusions

When analysing the effect of exercise and sport involvement, the meta-analysis showed that the mechanical stimuli provided by studies investigating either weight-bearing or non-weight-bearing movements (i.e., in a terrestrial and aquatic environments), evidenced the tendency to have a higher effect on BMD in the leg, pelvic, and lumbar bone sites of athletes, when compared to the bones of healthy non-athletes or non-active peers. However, the effect promoted by low-impact stimuli exercises or sports (e.g., practice in an aquatic environment) showed an overall tendency to be reduced when compared to the stimuli derived from impact and muscle tension of the practices in a terrestrial environment. Moreover, the effect of exercise and sport involvement on BMD was observed for both male and female participants of the studies. Therefore, exercise and sport involvement (e.g., long-term training) have potential to increase BMD beyond the healthy values reached by the young and middle-aged adult control group participants.

With regard to the effect of intervention planned with exercise and sport, the meta-analysis evidenced positive effects only on the BMD of the spine bone sites. There is evidence supporting that BMD alterations have a tendency to be high with long-term interventions (~12 months) for both sexes, and mainly when planned exercise and sport have a high muscle tension stimulus. Moreover, men (not women) exhibit an additional tendency to improve BMD after middle-term interventions (3 > months < 6) when strength training is planned.

Therefore, the current study supported the cause–effect of exercise and sport involvement or intervention on the values of BMD, evidencing that long-term involvement with exercise and sport plays a role in bone mass gain directly and independent of sex and bone health status. In addition, intervention including muscle tension stimuli (e.g., exercises with high-intensity load) enhances the effect on BMD after interventions lasting more than three and less than twelve months.

## Figures and Tables

**Figure 1 ijerph-20-06537-f001:**
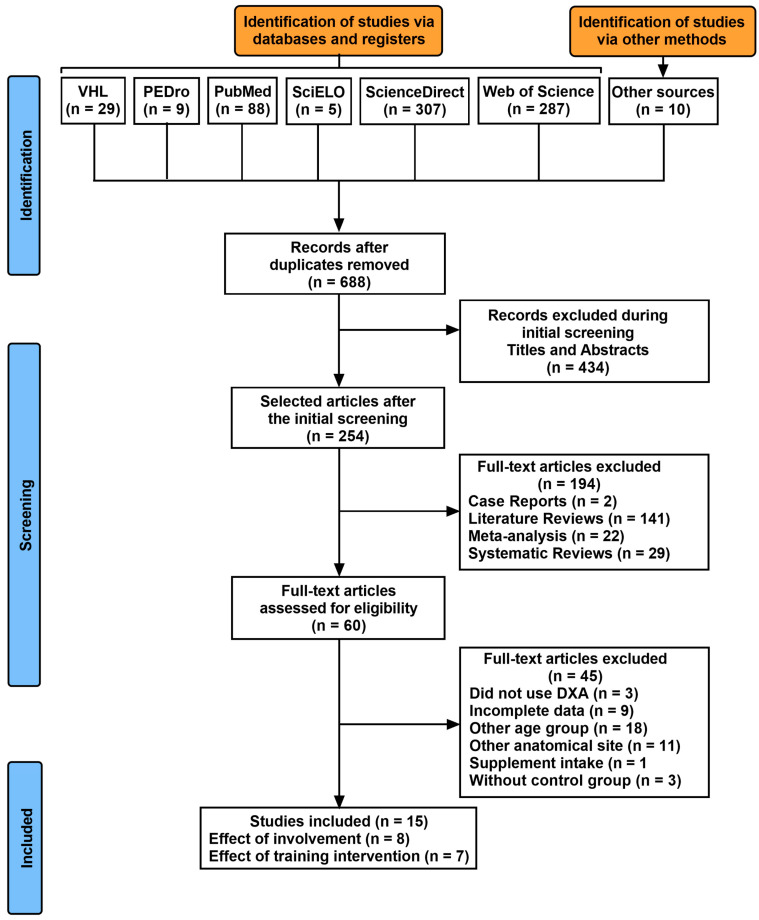
PRISMA study flow diagram.

**Figure 2 ijerph-20-06537-f002:**
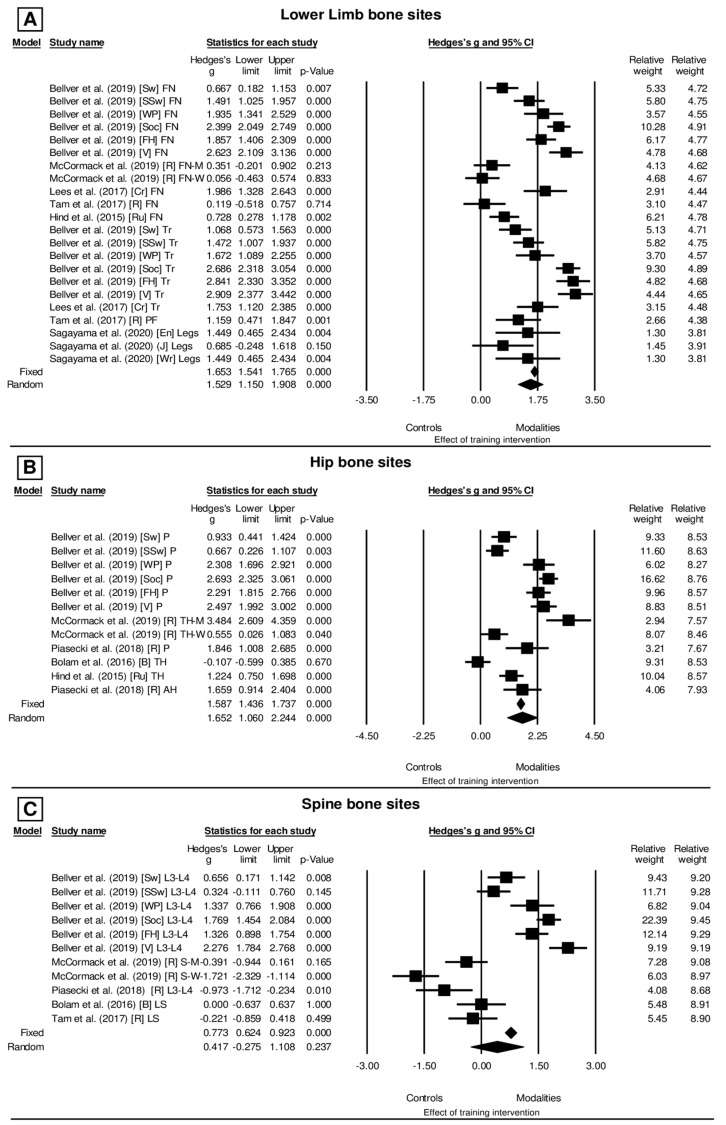
(**A**–**C**) Forest plot of the effect size for the values of BMD reported for each bone site, according to the exercises or sports analysed in the current study. Obs.: FN: Femoral Neck, Tr: Trochanter, PF: Proximal Femoral, P: Pelvis, TH: Total Hip, AH: Average Hip, L3-L4: Lumbar Vertebrae 3–4, LS: Lumbar Spine, Legs: Both Legs, S-M: Lumbar Man, S-W Lumbar Woman, Sw: Swimmers, SSw: Synchronized Swimmers, WP: Water Polo, S: Soccer, FH: Field Hockey, V: Volleyball, Cr: Cricket, R: Running, Ru: Rugby, B: Boxing, Wr: Wrestlers, J: Judo, En: Endurance Athletes. The markers represent the effect size and 95% CI.

**Figure 3 ijerph-20-06537-f003:**
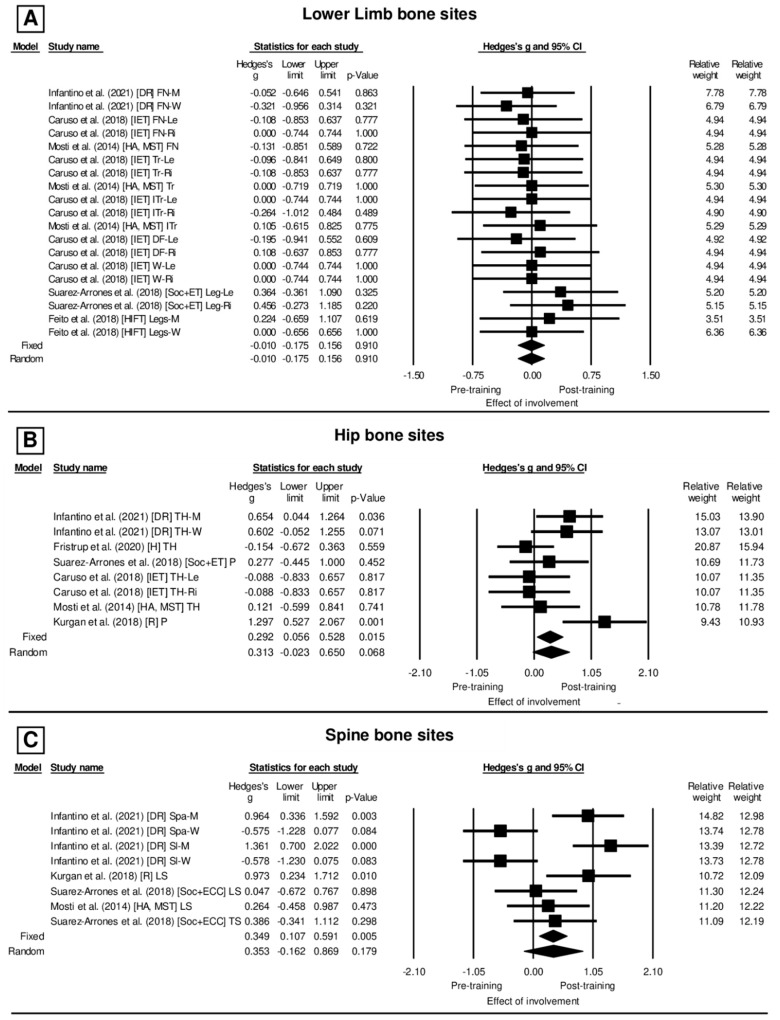
(**A**–**C**) Forest plot of the effect size for the BMD alterations in each bone site, according to the exercises or sports analysed in the current study: FN: Femoral Neck, FN-Le: Femoral Neck Left, FN-Ri: Femoral Neck Right, Tr-Le: Trochanter Left, Tr-Ri: Trochanter Right, Tr: Trochanter, ITr-Le: Inter Trochanter Left, ITr-Ri: Inter Trochanter Right, ITr: Inter Trochanter, DF-Le: Distal Femur Left, DF-Ri: Distal Femur Right, W-Le: Ward Triangle Left, W-Ri: Ward Triangle Right, Legs-W: Legs Women, Legs-M: Legs Men, P: Pelvis, TH-Le: Total Hip Left, TH-Ri: Total Hip Right, LS: Lumbar Spine, TS: Thoracic Spine, Spa: Anterior–Posterior Spine, SI: Lateral Spine, EIT: Inertial Exercise Trainer, HA: High Acceleration, MST: Maximal Strength Training, Soc: Soccer + ET: Eccentric-overload Training, HIFT: High Intensity Functional Training, IET: Inertial Exercise Trainer, R: Running, and DR: Distance Runners. The markers represent the effect size and 95% CI.

## Data Availability

The data that support the findings of this study are available from the corresponding and last author (mario.espada@ese.ips.pt and dalton.pessoa-filho@unesp.br) upon reasonable request.

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
