# Peer review of "Effects of Exercise and Sports Intervention and the Involvement Level on the Mineral Health of Different Bone Sites in the Leg, Hip, and Spine: A Systematic Review and Meta-Analysis"

_ijerph, 2023, doi:10.3390/ijerph20156537_

Round 1
Reviewer 1 Report
Although the authors have done a hudge job, the study has a lot of methodological issues that make the results /conclusions unreliable.
The title refers to the Mineral Health of Hip Bones Sites. However, the aim of the study is different in the abstract and the text (ln 83-86). Please clarify this.
1. If the aim of the study was to examine the BMD of the hip, many of the studies included in the analysis should be removed. Tables should be changed accordingly because many studies refer to spine and pelvis.
2. Moreover, even if the hip is the target of the present study, the anatomical sites studied are not the same in all studies included in the analysis. Other studies have studied the femoral Neck, the Trochanter, the Proximal Femoral, the Total Hip, or the Average Hip. Thus, the results are not comparable.
3. Was the BMD normal at baseline – can a normal be more normal? It is different if someone starts with lower BMD, the study of Denger et al. is a good example. Currently, osteoporosis and low bone mass are defined using T-scores; osteoporosis: T-scores of ≤ −2.5 and Low bone mass: −1.0 to −2.5, respectively.
4. Race/ethnicity and age are parameters that are not considered in the analysis. It is already found that the age of attainment for peak BMD at the hip is around 20 years. Moreover, menstrual cycle is a known factor affecting BMD especially in the female athletes. Since the athletic triad is a known problem in sports, the parameters “menstrual disorder and altered mineral bone density” should have been checked in the studies included.
5. Physical activity is different from exercise and the training characteristics have not been taken into account in the analysis. Especially, the years of training and the level of the athletes included. Healthy subjects that participated in a training program cannot be in the same analysis with professional athletes.
6. Why longitudinal and cross-sectional studies are only included. Why not RCTs?
7. Discussion should be focused on the research questions and not on the type of the studies. Please clarify the research questions and try to answer them. There is no clear clinical message, when the results are discussed according to whether the studies are cross-sectional or longitudinal.
Minor issues: The text included long sentences, even more than 6 lines. eg. 270-276, 283-289, 296-301, 402-408.
Author Response
My fellow authors and I would like to thank you for revising this manuscript, which comments have enriched this manuscript substantively. We believe we have adequately addressed each of the comments. However, if you deem that more changes are necessary, we look forward to addressing any other concerns. Please, find the response letter attached.

Reviewer 2 Report
This is a well written and well-structured meta-analysis in relation of bone mineral density and content considering physical activity and sports.
One suggestions to improve the introduction is to mention (briefly) how these parameters were assessed.
All the studies selected have athletes? (without taking into account control groups)
Besides, a short introduction of the methodology of the studies selected (type of sport, time of assessment, follow-ups) could be added in results section.
Discussion has a lot of ideas but mixed. Also, subsections could help the reader to understand all the information.
Maybe I am wrong, but Pedro is used for the quality assessment of clinical trials, do you think that another scale could be more suitable for this assessment?
Another limitation that could be added the lack of information about the differences between men and women.
Author Response

(The authors gave the same response as above.)

Reviewer 3 Report
Thank you for inviting me to review this manuscript. It is an interesting systematic review exploring effect of different exercise and sport modalities involvement on CMD. The authors should be congratulated on conducting this review, however to improve the manuscript I would suggest the following:
TITLE I would suggest revisiting the title to show that authors included terrestrial vs aquatic sport in adults. Also, I am unsure that the activities listed in the review could be considered physical activity. Physical activity is every movement that is carried out by the skeletal muscles that requires energy. While exercise is planned, structured, repetitive and intentional. I would suggest the research team to discuss and consider revisiting the title for more accurate information.
ABSTRACT
Please make sure to explain the BMD abbreviation the first time you use it (line 38 and not 41).
Conclusion of the abstract is quite difficult to understand. In particular, I suggest the authors to revisit the following sentence: The stimuli provided by impact and muscle tension, while performing sports and physical activities in a terrestrial environment, have a high effect on the BMD of the hip region bones compared to the BMD in non-and athletes involved with sports or physical activities providing low impact stimuli (such as..?). Please try to be more specific and detailed.
Also, I would suggest highlighting the role of metabolic health and medications as potential confounders.
INTRODUCTION
I would suggest adding information and more details on how the stimulus provided by PA and sport could have an effect on bones (i.e. evidence observed changes in muscle and bone muscle associated with PA (Hamrick, 2010) according to the mechanostat theory where muscle is recognised as source of mechanical stimuli for bone tissue (a decline in mechanical loading from muscle can reduce bone formation) (Edwards et al., 2015; Kaji, 2014, Sharir et al., 2011). Muscle and bone tissues also share a connection between humoral factors such as myokines (secreted from skeletal muscle cells) and osteokines (secreted from osteocytes) (Sharir et al., 2011). However, the precise mechanisms responsible for bone and skeletal mass metabolism are still not well-characterized. Recent data support the idea that muscle and bone tissue secrete factors targeting other tissues and are involved in glucose metabolism (Schafer et al., 2016).)
METHODS
Please clarify why the cut off of 45 years old was selected.
Please provide information on researchers involved in the screening and quality assessment. Were trained researchers? Which background they have?
Please provide information on time. When was the search conducted? The timeframe for the studies to be included which was? Please add more details on this.
It is not clear why the same quality assessment tool was used for cross sectional and longitudinal studies. Could the authors please clarify this? I would think that it is more appropriate to select different tools based on the type of study.
RESULTS
I would suggest the authors to add paragraphs describing briefly the different sports included in the review.
DISCUSSION
Please add limitations section for your systematic review.
I think the conclusion should be better supported by the analysis and results presented. Please make sure to adjust the sections above to support your message.
Please make sure to check spelling and grammar errors within the text.
Author Response

(The authors gave the same response as above.)

Round 2
Reviewer 1 Report
The authors have addressed all my comments. The paper in the present revised version can be published.
Reviewer 3 Report
Thanks for your answers, I am happy with the current version
Thanks for your answers, I am happy with the current version